# Predicting Bacteremia among Septic Patients Based on ED Information by Machine Learning Methods: A Comparative Study

**DOI:** 10.3390/diagnostics12102498

**Published:** 2022-10-15

**Authors:** Vivian Goh, Yu-Jung Chou, Ching-Chi Lee, Mi-Chia Ma, William Yu Chung Wang, Chih-Hao Lin, Chih-Chia Hsieh

**Affiliations:** 1Department of Emergency Medicine, National Cheng Kung University Hospital, College of Medicine, National Cheng Kung University, Tainan 70101, Taiwan; 2Clinical Medicine Research Center, College of Medicine, National Cheng Kung University, Tainan 70101, Taiwan; 3Department of Statistics and Institute of Data Science, College of Management, National Cheng Kung University, Tainan 70101, Taiwan; 4Waikato Management School, University of Waikato, Hamilton 3240, New Zealand

**Keywords:** bacteremia, blood culture, machine learning, logistic regression, net reclassification index

## Abstract

Introduction: Bacteremia is a common but life-threatening infectious disease. However, a well-defined rule to assess patient risk of bacteremia and the urgency of blood culture is lacking. The aim of this study is to establish a predictive model for bacteremia in septic patients using available big data in the emergency department (ED) through logistic regression and other machine learning (ML) methods. Material and Methods: We conducted a retrospective cohort study at the ED of National Cheng Kung University Hospital in Taiwan from January 2015 to December 2019. ED adults (≥18 years old) with systemic inflammatory response syndrome and receiving blood cultures during the ED stay were included. Models I and II were established based on logistic regression, both of which were derived from support vector machine (SVM) and random forest (RF). Net reclassification index was used to determine which model was superior. Results: During the study period, 437,969 patients visited the study ED, and 40,395 patients were enrolled. Patients diagnosed with bacteremia accounted for 7.7% of the cohort. The area under the receiver operating curve (AUROC) in models I and II was 0.729 (95% CI, 0.718–0.740) and 0.731 (95% CI, 0.721–0.742), with Akaike information criterion (AIC) of 16,840 and 16,803, respectively. The performance of model II was superior to that of model I. The AUROC values of models III and IV in the validation dataset were 0.730 (95% CI, 0.713–0.747) and 0.705 (0.688–0.722), respectively. There is no statistical evidence to support that the performance of the model created with logistic regression is superior to those created by SVM and RF. Discussion: The advantage of the SVM or RF model is that the prediction model is more elastic and not limited to a linear relationship. The advantage of the LR model is that it is easy to explain the influence of the independent variable on the response variable. These models could help medical staff identify high-risk patients and prevent unnecessary antibiotic use. The performance of SVM and RF was not inferior to that of logistic regression. Conclusions: We established models that provide discrimination in predicting bacteremia among patients with sepsis. The reported results could inspire researchers to adopt ML in their development of prediction algorithms.

## 1. Introduction

Bacteremia is a common healthcare problem encountered by clinicians, with a community incidence of approximately 0.82% [1]. Despite the development of therapeutic strategies and antimicrobial therapy, bacteremia is usually considered a life-threatening infectious disease if organ dysfunction occurs [2]. Therefore, prompt administration of appropriate antimicrobials remains the cornerstone of bacteremia treatment to achieve favorable prognoses [3]. Generally, the gold standard for diagnosis of bloodstream infections is microbial growth on blood cultures [4]. However, microbial growth is time-consuming, and the result cannot be recognized immediately by first-line physicians in the emergency department (ED) [5]. Therefore, ED physicians usually need to decide whether to order a blood culture and prescribe empirical antibiotics to patients when they suspect bacteremia based on their experience and intuition without microbiologic support [6,7].

The clinical presentation of bacteremia varies and largely depends on the infection site, host immune status or comorbidities, causative microorganisms, and severity of illness at onset [8]. However, because a well-defined rule to assess patient risk of bacteremia and the urgency of blood culture is lacking for ED physicians [9], the incidence of bacteremia is often underestimated, and excessive culture examinations are conducted [5]. We believe the underestimation of bacteremia in patients could result in the delayed administration of appropriate antibiotics, and excessive culture examination could result in a waste of medical resources [10] and expose patients to unnecessary risks [5].

To prevent unnecessary blood culture examination, numerous studies aiming to identify populations with a high risk of bacteremia have been reported [11]. However, the utilization and validation of these studies are often limited [11]. The majority of models can only be applied to specific populations [12] or specific infectious foci [13,14], and some models achieve poor discrimination in bacteremic patients [15]. To date, numerous reports have addressed ED patients to predict the occurrence of bacteremia, but their targeted populations were patients with infections suspected by ED physicians [16,17]. Investigations detailing septic population to establish a valuable model for prediction of bacteremia are lacking in clinical ED practice.

The disposition of large-scale information has been realized through the introduction of information technologies and electronic medical records (EMRs) at modern medical institutes. Patient physiologic and laboratory parameters are often extracted to build predictive models using machine learning (ML) [18]. Several studies have proposed the prediction of bacteremia using multilayer perceptron, random forest (RF), and gradient boosting algorithms [19] or logistic regression (LR) and support vector machines (SVM) via machine learning techniques [20]. However, few analyses have compared machine learning methods. Based on the advantage of ML in managing big data [21], the aim of this study is to establish a predictive model for bacteremia in septic patients using big data available in the ED through LR, SVM, and RF.

## 2. Material and Methods

### 2.1. Study Design and Data Collection

We conducted a cohort study consisting of retrospectively captured target patients in the ED of National Cheng Kung University Hospital (a university-affiliated medical center with 1400 beds) from January 2015 to December 2019. The inclusion criteria were ED adults (≥18 years) presenting with systemic inflammatory response syndrome (SIRS) who had blood cultures during their ED stay. Exclusion criteria included patients with incomplete information detailing our measurements and outcomes. The layer sampling method was used to divide all the targeted patients into a 70% derivation dataset and a 30% validation dataset.

### 2.2. Measurements and Outcomes

All variables available in the ED were captured for analyses regarding patient demographics, previously identified comorbidities, vital signs upon ED triage, and laboratory data. Past comorbidities were retrieved from EMRs at the hospital and were determined by the international statistical classification of diseases and related health problems 10th revision codes [22]. Vital signs included body temperature, heart rate, respiratory rate, blood pressure, oxygen saturation, and Glasgow coma scale. Similarly to a previous investigation [5], laboratory information was collected within 12 h after ED arrival. Laboratory parameters included white blood cell count and platelet count. Each parameter of vital signs and laboratory measurements was transferred to the categorical variable for further analysis. The primary outcome was the occurrence of bacteremia diagnosed in the ED. Two investigators, one board-certified emergency medicine physician and an infectious disease clinician, independently reviewed the computerized records. Any reviewing discrepancy was resolved by discussion between the investigators in periodic meetings.

### 2.3. Definition

True bacteremia is defined as the causative pathogen yielded in at least one blood culture after excluding contaminated sampling [23]. The growth of potentially contaminated pathogens on blood cultures includes coagulase-negative staphylococci, *Clostridium perfringens, Micrococcus* spp., *Bacillus* spp., *Propionibacterium* spp., and Gram-positive bacilli according to previously published criteria [16]. Patients with contamination were regarded as having no bacteremia episodes and needing further analysis. Patients with SIRS were recognized based on the Sepsis-2 criteria [24]; information about patients’ comorbidity was retrieved from EMRs and recorded with specific International Classification of Diseases 10 (ICD-10) codes [22] (shown in Appendix A).

### 2.4. Ethical Considerations

This study was approved by the Institutional Review Board of National Cheng Kung University Hospital (B-ER-111-064). The requirement for informed consent was waived because the captured information was deidentified prior to analysis.

### 2.5. Statistical Analysis

The Statistical Package for the Social Sciences for Windows version 23.0 (Chicago, IL, USA) was used for descriptive statistical analyses; R version 4.1.2 packages were used for LR, SVM, and RF methods; and the significance level in this study was set to 0.01. Continuous variables were described as the median (interquartile range, IQR) or mean (standard deviation, SD), and categorical variables were expressed as numbers (percentages). Categorical variables were compared using the Pearson *chi*-square test, and continuous variables were adopted for independent t-tests. For the assessed models, the area under the receiver operating characteristic curve (AUROC) was applied to assess performance in differentiating bacteremic patients from non-bacteremic populations. The model with the highest AUROC value was chosen for further comparison with the ML algorithms.

### 2.6. Logistic Regression

The LR model is a linear regression with the primary purpose of establishing the relationship between the binary response variable, such as whether the event occurred, and the explanatory variables [25]. The mathematical equation of LR is expressed as follows.
$$log\left(\frac{p}{1-p}\right)=\beta_{0}+\beta_{1}X_{1}+\beta_{2}X_{2}+\cdots \ldots +\beta_{m}X_{m}$$
where p is the probability of an event occurring, $\frac{p}{1-p}$ is the odds or risk for very low p values, $X_{i}$ is the ith explanatory variable, and $\beta_{i}\;$is the coefficient of $X_{i}$. Before model building, screening and selection of adequate explanatory variables were necessary for LR. Excessive variables would calculate the regression coefficient complex [26]. The methods of forwarding selection and increasing interaction terms were applied to separately select the adequate variables.

The forward selection method was used to build model I as a stepwise regression. First, variables were added to the model one by one. In each forward step, the variable that best improved the model was picked up and added to the model. Next, the interaction term was applied to establish model II. The method was used to create new variables representing the interactions between the existing variables. The interactions between the variables were subjectively judged based on the operator’s experience and knowledge in an attempt to improve the performance of the model.

ROC-AUC and Akaike information criterion (AIC) were adopted for these two LR-built models to determine which model was superior. The AIC was used to measure the models’ complexity and the goodness of fit; the lower the AIC, the better the model [27].

### 2.7. Support Vector Machine and Random Forest

SVM is a machine learning model commonly used to deal with classification problems. SVM is a linear classifier that can be used to solve issues such as small samples, nonlinearity, and high dimensionality. The main concept is to construct a hyperplane to separate and classify sample data. This hyperplane correctly separates the two types of samples and maximizes the distance between the two groups. When encountering nonlinear problems, the data that cannot be linearly classified in low dimensions can be projected into high-dimensional space using kernel function transformation. Then, the data can be organized by establishing a hyperplane. SVM has a positive effect on classification problems, so it has been widely used in classification problems in various fields in recent years [28].

RF is a machine learning model that consists of multiple decision trees. The decision tree includes the root, parent, child, and leaf nodes. It is an analysis method of tree structure used to deal with classification problems. A general decision tree starts from the root, branches with features, is divided into two or more child nodes, and continues to branch until the self-defined stopping condition, that is, to the leaf. Each internal node uses a feature branch, each branch represents a possible field output outcome, and each endpoint represents the final predicted or decided result of a given classification. RF can handle classification and regression tasks, and used features can be discrete or continuous data. Furthermore, because RF uses random sampling and the selection of features to construct multiple decision trees in the operation process, it can reduce the occurrence of overfitting [29].

Because the true positivity rate of bacteremia in the original data was 7.7%, they was imbalanced data. Imbalanced data can significantly compromise the distributive characteristics of most standard learning algorithms and ultimately result in unfavorable predictive accuracy [30]. Therefore, data processing was necessary before model building. First, we dealt with the training dataset using the methods of oversampling, undersampling, and random oversampling (ROSE). Then, the adjusted datasets were used to build model III with the SVM algorithm and model IV with the RF algorithm. A total of 500 trees were generated, and their depth was 5 in model built with RF.

### 2.8. Net Reclassification Index

These models were compared with the net reclassification index (NRI) to determine which model was superior [31]. If the value of NRI was more than 0 and the result was statistically significant, the comparison model was better than the original model.

The null hypothesis of this test is that “NRI ≦ 0” because the equation of the test statistic (*z*) is:$$z=\frac{\hat{NRI}}{\sqrt{\frac{{\hat{P}}_{p,up}+{\hat{P}}_{p,down}}{N_{bacteremia}^{2}}+\frac{{\hat{P}}_{n,up}+{\hat{P}}_{n,down}}{N_{non-bacteremia}^{2}}}}$$

${\hat{P}}_{p,up}$ is the possibility that patients with bacteremia were predicted as nonbacteremic in the original model and as bacteremia in the comparison model.

${\hat{P}}_{p,down}$ is the possibility that patients with bacteremia were predicted to have bacteremia in the original model and non-bacteremia in the comparison model.

${\hat{P}}_{n,up}$ is the possibility that patients without bacteremia were predicted to have bacteremia in the original model and non-bacteremia in the comparison model.

${\hat{P}}_{n,down}$ is the possibility that patients without bacteremia were predicted as nonbacteremic in the original model and as bacteremia in the comparison model.

$N_{bacteremia}$ is the number of patients with bacteremia, and $N_{non-bacteremia}$ is the number of patients without bacteremia.

The level of significance was 0.01. If the value of *z* was above 2.326, then the null hypothesis was rejected. This result suggests that the comparison model was superior to the original model.

## 3. Results

### 3.1. Study Population

During the study period, 437,969 patients visited the study ED. Blood cultures indicated that 41,416 of these adults had SIRS; 40,395 were enrolled as the targeted cohort after excluding 312,836 SIRS < 2, 69,338 not receiving blood culture, 14,379 patients younger than 18 years old, and 1021 with incomplete information (Figure 1). Patients with bacteremia episodes in the ED accounted for 7.7% (4197 patients) of the targeted cohort. The demographic and clinical characteristics of the bacteremic and non-bacteremic patients are shown in Table 1. Compared to patients without bacteremia episodes, those with bacteremia were older and more frequently had specific comorbidities, namely diabetes, liver disease, chronic kidney disease, and malignancy. There were also significant differences in white blood cell count, platelet count, body temperature, heart rate, respiratory rate, blood pressure, and consciousness level. The patient demographics and clinical characteristics were similar between the derivative and validation patients (Figure 2).

### 3.2. Model Training with Logistic Regression

For the derivation dataset, the significant variables of patients enrolled in the logistic regression model (i.e., models I and II) are shown in Table 2. The AUC in models I and II were 0.729 (95% CI, 0.718–0.740) and 0.731 (95% CI, 0.721–0.742), with AICs of 16,840 and 16,803 (Table 3), respectively. Owing to the correlation between patient age and uncomplicated diabetes, the interaction term of age and diabetes mellitus was added in model II. Consequently, regardless of which dataset (derivation or validation) was used, the performance of model II was superior to that of model I. Figure 3 shows a diagram of the interaction between age and uncomplicated DM. When uncomplicated DM = 1, log(odds) = −3.505 + 0.004 × age. The amount of increased risk is 0.004 as the age of the patient increases by one year. When uncomplicated DM = 0, log(odds) = −5.12 + 0.024 × age. The amount of increased risk is 0.024 as the age of the patient increases by one year. Risk increased more rapidly in patients without uncomplicated DM than in patients with uncomplicated DM.

### 3.3. Model Training with Support Vector Machine and Random Forest

The performance of models III and IV in the derivation and validation of patients is shown in Figure 4 and Table 3, respectively. The AUC values of models III and IV in the derivation group were 0.751 (95% CI, 0.740–0.761) and 0.835 (95% CI, 0.825–0.844), and those in the validation dataset were 0.730 (95% CI, 0.713–0.747) and 0.705 (0.688–0.722), respectively. The performance of model III in predicting bacteremia patients was inferior to that of model IV in the derivation patients, but the superiority of model III was exhibited in validation patients. Notably, the performance of the SVM and RF models was discrepant in the derivation and validation datasets.

### 3.4. Comparison of Support Vector Machine, Random Forest, and Logistic Regression

We used the ROC curve and the tangent of slope 1 to solve the sensitivity, specificity, and confusion matrix. The sensitivity, specificity, and positive likelihood ratio each of the models were 0.660, 0.653, and 1.902 for LR; 0.698, 0.631, and 1.892 for SVM; and 0.661, 0.639, and 1.831 for RF. The LR model with the best performance (model II) was tested with SVM and RF models. However, as shown in Table 4, irrespective of the SVM or RF model, there was no statistical evidence to support that the performance of the model created with LR was superior to that of the models created with SVM and RF.

LR and RF models have similar significant variables, including age, gender, chronic obstructive pulmonary disease, uncomplicated diabetes, hemato-oncology, white blood cell > 12,000/μL, band cells > 10%, platelet < 140,000/μL, body temperature, and heart rate. Mean arterial pressure, respiratory rate, and Glasgow coma scale were considered significant in the RF model but not in the LR model (Table 5).

## 4. Discussion

Based on clinical information available in the ED, we established algorithms with proper discrimination in predicting bacteremia episodes among septic patients. We hope that the reported results will inspire ED physicians to develop a prediction model to manage the clinical problems they face daily. Furthermore, the SVM and RF models predicted bacteremia with similar effectiveness as those established by LR. The advantage of the SVM and RF models is that the prediction model is more elastic and not limited to a linear relationship. The advantage of the LR model is that it is easy to explain the influence of the independent variable on the response variable. Each method has its advantages. Because the performance of the SVM and RF models is equal to that of the traditional LR model, medical researchers should be open-minded about adopting SVM or RF for the development of prediction algorithms.

Based on patient demographics and laboratory data available in the ED, the algorithms with proper discriminations in predicting bacteremia episodes among septic patients were evident. Similar to previous studies indicating that the performance of SVM and RF models was not inferior to that of traditional LR models [32,33,34], the prediction performance of the SVM and RF models was not inferior to that of the LR-based model in our study.

Sepsis is a life-threatening infectious disease resulting in organ dysfunction caused by dysregulated host immunity [35]. Sepsis is a common healthcare problem encountered by clinicians because it is a heterogeneous syndrome with varying clinical presentations and characteristics [36]. In addition, the therapeutic outcomes of septic patients differ depending on socioeconomic status, location of episodes, host immune status or comorbidities, causative microorganisms, sites of infection, the severity of illness at onset, and quality of care [37,38,39]. In 2016, the Sepsis-3 criteria proposed the quick Sequential (sepsis-related) Organ Failure Assessment (qSOFA) as a replacement for the SIRS score, issued by previous Sepsis-2 measures [40] for the early screening of sepsis outside of intensive care units because SIRS scores were deemed to have unsatisfactory specificity and sensitivity in detecting septic patients [41]. According to the Sepsis-3 criteria, bacteremia patients with initial qSOFA scores of ≥2 at ED arrival were identified early as septic candidates, and those with organ dysfunction (i.e., an increase in SOFA scores of ≥2 from the baseline score within three days of ED arrival) during hospitalization were identified as septic patients. However, this revised definition of sepsis syndrome is unsuitable for ED physicians. In addition, a previous bacteremia investigation indicated that the contemporary definition is unsafe in EDs before culture information on bacteremia is recognized [42]. Accordingly, in the present study, we included patients with sepsis who met the aforementioned Sepsis-2 criteria as the target population.

Although blood culture study is considered the gold standard for diagnosis of bacteremia, false-positive results might be misinterpreted and result in patient harm. The probability of true positive blood culture, which means that the causative pathogen is identified, is low. The rate ranges from 4.1% to 7% [8,9,43]. The low yield rate of blood culture study represents a financial burden for hospital laboratories [44]. On the other hand, false-positives influenced by contamination occur at a similar or even higher rate [9,10,45]. A general blood culture study without patient selection might not be beneficial and could be harmful to the patients and medical staff. False positives could subject patients to unnecessary antibiotic treatment and hospitalization, harming patients and increasing the burden on healthcare workers. Our model may help medical staff identify high-risk patients, decrease the amount of unnecessary blood culture studies, and reduce the incidence of false-positive results. Both patients and the hospitals can benefit from our proposed model.

The medical community has widely used the method of logistic regression for the development of prediction models [46]. LR was first proposed in 1958. It is a kind of linear regression helpful for the analysis of the correlation between explanatory and response variables [25]. However, LR is based on theory and linear assumptions. It is usually limited by human intervention and subjective knowledge and results in a lack of flexibility [47]. Benefiting from the development of EMRs, SVM and RF have been gradually adopted as tools to exploit clinical risk prediction models [21,48,49]. Compared to LR models, SVM and RF models have higher flexibility as a result of including nonlinear association and interaction terms [50]. In addition, SVM and RF models perform exceptionally well when dealing with multiple variables [48,51].

On the other hand, the modelling of SVM or RF is sometimes so complex that humans cannot straightforwardly interpret it. This condition is the so-called “black box” [52]. The black box is a significant concern for many clinicians in adopting an algorithm-based aspect in their research [53,54,55]. Notably, some studies have suggested that the critical variables of SVM and RF models are usually consistent with clinical intuition and significant predictors found in prior studies in the field [32,33,34]. Accordingly, the essential variables chosen in the RF model are primarily compatible with the variables from the LR model presented in our study. However, variables that are distinct from those of LR models (e.g., mean arterial pressure, respiratory rate, and Glasgow coma scale) have previously been found to be relevant to bacteremia [5,16,56]. Therefore, the variables inside the black box may not be too ambiguous to be interpreted. Consistent with previous studies indicating that the performance of SVM and RF models is not inferior to that of traditional LR models [32,33,34], our findings reveal that SVM or RF should not be a barrier for clinicians in predicting bacteremia episodes in septic ED patients. Recent worldwide advances in EMR have created a suitable environment to leverage SVM and RF to improve the quality of patient care. Therefore, we believe that now is a good time to integrate disparate data sources through SVM or RF to achieve real-time decisions. Beside SVM and RF, deep learning is a subset of machine learning that completely relies on artificial neural networks. This learning machine is input with raw data and establishes its own representation required for pattern recognition. The revolution of deep learning can aid in optimizing pathways of diagnosis and prognosis to develop individualized treatment plans. This field had achieved promising results in the image and language sector since the digitization of medical records. In this manner, machine learning systems could represent an opportunity for medical providers to benefit from studies that require large datasets, such as multicohort studies or object classification in future studies.

In the current study, the performances of SVM and RF models in predicting bacteremia patients were dissimilar between the derivation and validation datasets. We believe the following were the leading reasons contributing to this finding. First, the derivation dataset was imbalanced, and data processing was needed prior to model training. Therefore, oversampling, undersampling, and ROSE were adopted to deal with the data before model establishment. Thus, there were three subgroups of the derivation dataset with different methods of prior data processing. Consequently, all these subgroup datasets were used to develop models in the derivation dataset and then examined in the validation dataset. In our opinion, the methods for establishing model IV are reasonably appropriate for management of clinical information.

## 5. Limitations

Our study is subject to several limitations. First, this is a single-center retrospective study; unfortunately, we do not have complete data on patients with no prior visits, except for age and sex. Therefore, diagnosing sepsis or bacteremia mainly depends on coding by ED physicians, leading to information bias when analyzing the outcome. Second, confined to the dataset of a single medical center, external validation with a dataset from other medical institutes is required to improve the predictive power and accuracy of the proposed model and for potential broader utilization. Third, to reduce categorization bias, all clinical information was randomly retrieved by two physicians, who inspected medical records together to solve discrepancies. Finally, the model was limited to laboratory analysis. Further integration and connection with existing information systems in the hospital are needed to develop the model as an automated decision support tool for ED practice. Balancing the model’s certainty and financial factors during development would be challenging.

## 6. Conclusions

Bacteremia is substantially associated with high morbidity and mortality, and prompt identification and intervention can vastly improve the survival of patients with bacteremia. Through clinical information captured in the ED, we established algorithms with useful discrimination in predicting bacteremia episodes among septic patients. Furthermore, the similar performance of the ML model and traditional logistic regression models in predicting bacteremia was established herein. Accordingly, medical researchers should be open-minded about adopting ML for the development of prediction algorithms to solve their clinical problems. We believe that our reported results will inspire ED physicians to develop a useful prediction model to manage the clinical problems they face daily.

## Figures and Tables

**Figure 1 diagnostics-12-02498-f001:**
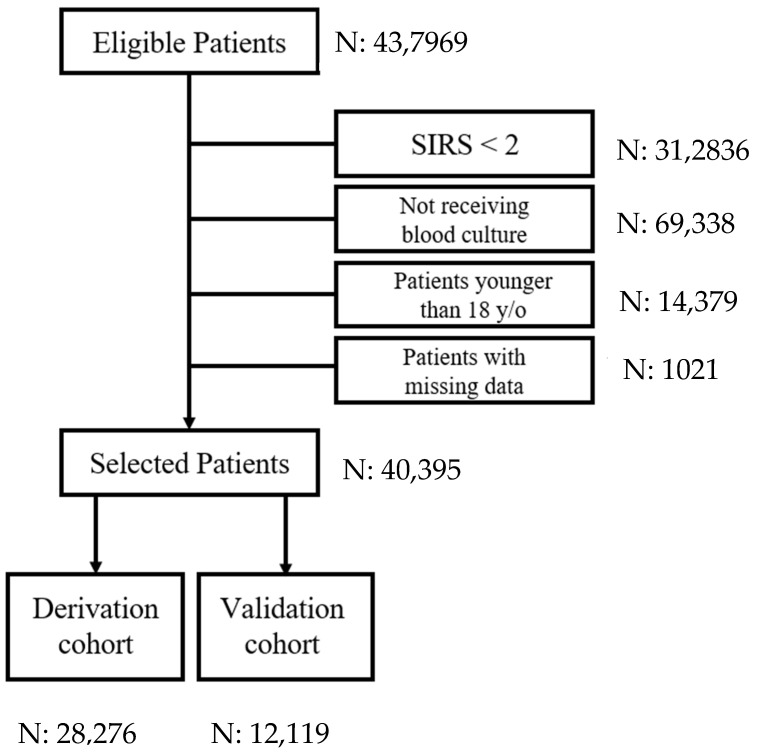
The selection process for the targeted cohort. (SIRS = systemic inflammatory response syndrome).

**Figure 2 diagnostics-12-02498-f002:**
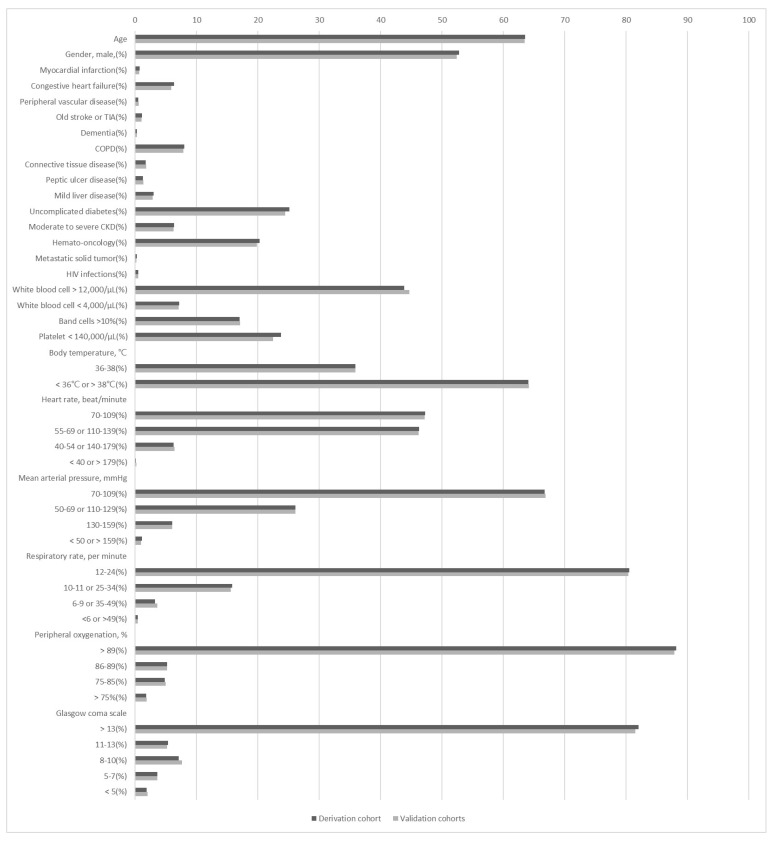
Patient demographics and laboratory data were similar between the derivation and validation cohorts. The *p* value of each variable was 0.999. CKD = chronic kidney disease; COPD = chronic obstructive pulmonary disease; HIV = human immunodeficiency virus; TIA = transient ischemic accident.

**Figure 3 diagnostics-12-02498-f003:**
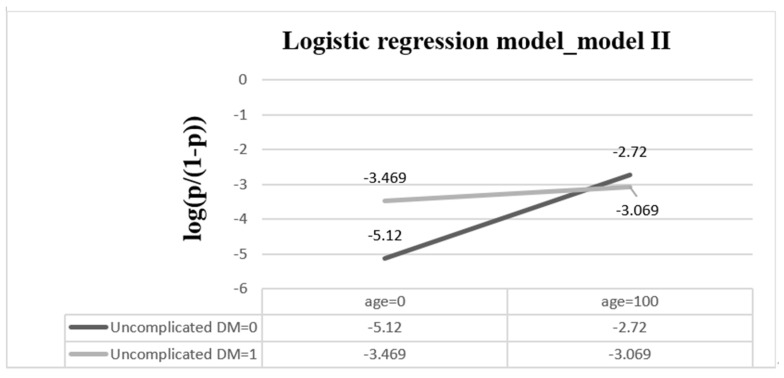
**The interaction of age and uncomplicated DM when other explanatory variables are based on****reference groups.** The equation of the logistic regression model is log(p/(1 − p)) *=* −5.12 + 0.024 × age + 1.615 × uncomplicated DM *−* 0.02 *×* age × uncomplicated DM. (DM = diabetes mellitus).

**Figure 4 diagnostics-12-02498-f004:**
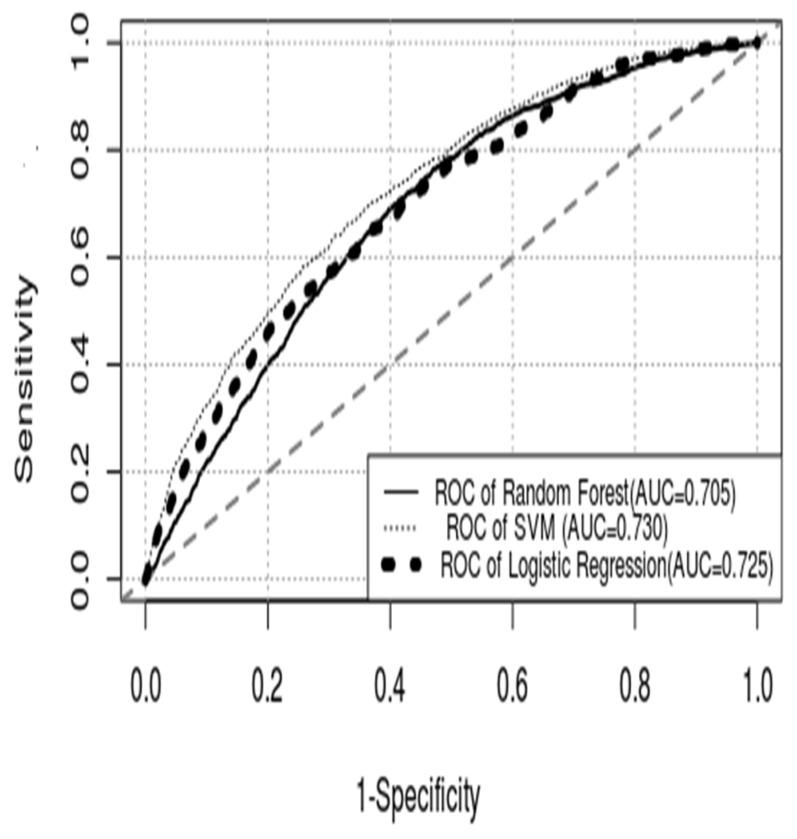
**ROC curve of each model.** (ROC curve = receiver operating characteristic curve; AUC = area under the curve; SVM = support vector machine).

**Table 1 diagnostics-12-02498-t001:** Differences in patient demographics and laboratory data between bacteremic and non-bacteremic patients.

Variables	Patient Number (%)						*p* Value
	All (*n* = 40,395)		Bacteremia (*n* = 4058)		Non-Bacteremia (*n* = 36,337)		
**Age, mean ± SD**	**63.5 ± 19.4**		**69.2 ± 15.4**		**62.9 ± 19.7**		<0.001
Gender, male	21,272	(53)	2032	(50)	19,240	(53)	<0.001
Comorbidities							
Myocardial infarction	289	(0.7)	23	(0.6)	266	(0.7)	0.277
Congestive heart failure	2502	(6.2)	231	(5.7)	2271	(6.3)	0.173
Peripheral vascular disease	231	(0.6)	25	(0.6)	206	(0.6)	0.776
Old stroke or TIA	440	(1.1)	53	(1.3)	387	(1.1)	0.186
Dementia	130	(0.3)	19	(0.5)	111	(0.3)	0.112
COPD	3220	(8)	130	(3.2)	3090	(8.5)	<0.001
Connective tissue disease	697	(1.7)	54	(1.3)	643	(1.8)	0.049
Peptic ulcer disease	521	(1.3)	57	(1.4)	464	(1.3)	0.542
Mild liver disease	1190	(3)	217	(5.4)	973	(2.7)	<0.001
Uncomplicated diabetes	10,078	(25)	1289	(32)	8789	(24)	<0.001
Moderate to severe CKD	2552	(6.3)	345	(8.5)	2207	(6.1)	<0.001
Hemato-oncology	8131	(20)	979	(24)	7152	(20)	<0.001
Metastatic solid tumor	108	(0.3)	8	(0.2)	100	(0.3)	0.451
HIV infections	206	(0.5)	16	(0.4)	190	(0.5)	0.330
Laboratory data in the ED							
White blood cell > 12,000/μL	17,812	(44)	2054	(51)	15,758	(43)	<0.001
White blood cell < 4000/μL	2900	(7.2)	389	(9.6)	2511	(6.9)	<0.001
Band cells > 10%	6884	(17)	1437	(35)	5447	(15)	<0.001
Platelet < 140,000/μL	9458	(23)	1588	(39)	7870	(22)	<0.001
Vital signs upon ED triage							
Body temperature, °C							<0.001
36–38	14,493	(36)	1108	(27)	13,385	(37)	
<36 °C or >38 °C	25,902	(64)	2950	(73)	22,952	(63)	
Heart rate, beats/minute							<0.0001
70–109	19,089	(47)	1735	(43)	17,354	(48)	
55–69 or 110–139	18,681	(46)	1903	(47)	16,778	(46)	
40–54 or 140–179	2550	(6.3)	405	(10)	2145	(5.9)	
<40 or >179	75	0.2	15	(0.4)	60	(0.2)	
Mean arterial pressure, mmHg							<0.001
70–109	26,962	(67)	2697	(66)	24,265	(67)	
50–69 or 110–129	10,547	(26)	1103	(27)	9444	(26)	
130–159	2444	(6.1)	188	(4.6)	2256	(6.2)	
<50 or >159	442	(1.1)	70	(1.7)	372	(1)	
Respiratory rate per minute							
12–24	32,496	(80)	3311	(82)	29,185	(80)	0.091
10–11 or 25–34	6357	(16)	585	(14)	5772	(16)	
6–9 or 35–49	1364	(3.4)	141	(3.5)	1223	(3.4)	
<6 or >49	178	(0.4)	21	(0.5)	157	(0.4)	
Peripheral oxygenation, %							0.006
>89	34,359	(88)	3405	(87)	30,954	(88)	
86–89	2026	(5.2)	205	(5.2)	1821	(5.2)	
75–85	1899	(4.9)	223	(5.7)	1676	(4.8)	
>75%	716	(1.8)	91	(2.3)	625	(1.8)	
Glasgow coma scale							<0.0001
>13	33,069	(82)	3097	(76)	29,972	(82)	
11–13	2150	(5.3)	278	(6.9)	1872	(5.2)	
8–10	2936	(7.3)	388	(9.6)	2548	(7)	
5–7	1450	(3.6)	176	(4.3)	1274	(3.5)	
<5	790	(2)	119	(2.9)	671	(1.9)	

CKD = chronic kidney disease; COPD = chronic obstructive pulmonary disease; HIV = human immunodeficiency virus; SD = standard deviation; TIA = transient ischemic accident. Data are expressed as numbers (%) unless indicated specifically, and boldface indicates statistical significance.

**Table 2 diagnostics-12-02498-t002:** The logistic regression method in the derivation dataset established significant variables in models I and II.

Variables	Model I			Model II		
	Odds Ratio	95% CI	*p* Value	Odds Ratio	95% CI	*p* Value
Age, years	1.021	1.018–1.023	<0.001	1.024	1.022–1.027	<0.001
Sex, male	0.902	0.832–0.978	0.012	0.887	0.818–0.962	0.003
Comorbidities						
Chronic obstructive pulmonary disease	0.407	0.327–0.506	<0.001	0.407	0.327–0.507	<0.001
Mild liver disease	1.624	1.34–1.967	<0.001	1.604	1.324–1.944	<0.001
Uncomplicated diabetes	1.253	1.146–1.37	<0.001	5.21	3.331–8.148	<0.001
Moderate to severe CKD	1.34	1.154–1.555	<0.001	1.32	1.137–1.532	<0.001
Hemato-oncology	1.155	1.048–1.272	0.004	1.152	1.046–1.27	0.004
Laboratory data in the ED						
White blood cell > 12,000/μL	1.598	1.464–1.745	<0.001	1.598	1.464–1.745	<0.001
White blood cell < 4000/μL	1.298	1.114–1.511	<0.001	1.299	1.115–1.513	<0.001
Band cell > 10%	2.847	2.607–3.109	<0.001	2.843	2.603–3.106	<0.001
Platelet < 140,000/μL	2.161	1.973–2.367	<0.001	2.151	1.964–2.357	<0.001
Vital signs upon ED triage						
Body temperature < 36 °C or > 38 °C	1.835	1.674–2.012	<0.001	1.85	1.688–2.028	<0.001
Heart rate, beats/minute						
55–69 or 110–139	1.279	1.174–1.394	<0.001	1.28	1.175–1.395	<0.001
40–54 or 140–179	2.025	1.748–2.346	<0.001	2.026	1.749–2.348	<0.001
<40 or >179	2.406	1.187–4.878	0.015	2.465	1.213–5.012	0.013
Age plus uncomplicated diabetes	-	-	-	0.98	0.974–0.986	<0.001

CKD = chronic kidney disease; CI = confidence interval; ED = emergency department.

**Table 3 diagnostics-12-02498-t003:** AUROC, AIC, and 95% confidence interval in the derivation and validation datasets.

Model	Algorithm	Derivation Dataset			Validation Dataset		
		AUROC	AIC	95% CI	AUROC	AIC	95% CI
I	Logistic regression	0.729	16,840	0.718–0.740	0.722	-	0.705–0.739
II	Logistic regression	0.731	16,803	0.721–0.742	0.725	-	0.708–0.742
III	Support vector machine	0.751	-	0.740–0.761	0.730	-	0.713–0.747
IV	Random forest	0.835	-	0.825–0.844	0.705	-	0.688–0.722

AUROC = area under the operating characteristic curve; AIC = Akaike information criterion.

**Table 4 diagnostics-12-02498-t004:** Comparison of the model II and machine learning models in the discrimination of bacteremia in the validation dataset.

Comparing Model	Net Reclassification Index	*p* Value
Model III (support vector machine)	−0.02	>0.01
Model IV (random forest)	0.01	>0.01

**Table 5 diagnostics-12-02498-t005:** Comparison of significant variables in the logistic regression and random forest models.

Model	Logistic Regression	Random Forest
Variables	Age	Age
	Gender	Gender
	COPD	COPD
	Uncomplicated DM	Uncomplicated DM
	Hemato-oncology	Hemato-oncology
	WBC > 12,000/μL	WBC > 12,000/μL
	Band cell > 10%	Band cell > 10%
	Platelet < 140,000/μL	Platelet < 140,000/μL
	Body temperature	Body temperature
	Heart rate	Heart rate
	Mild liver disease	Mean arterial pressure
	Moderate to severe CKD	Respiratory rate
	WBC < 4000/μL	Glasgow coma scale
	Age plus uncomplicated DM	

COPD = chronic obstructive pulmonary disease; DM = diabetes mellitus; WBC = white blood cell; CKD = chronic kidney disease.

## Data Availability

Data available on request due to restrictions of ethical.

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
