# Peer review of "Predicting Bacteremia among Septic Patients Based on ED Information by Machine Learning Methods: A Comparative Study"

_diagnostics, 2022, doi:10.3390/diagnostics12102498_

Round 1
Reviewer 1 Report
The authors compared some supervised machine learning methods in predicting bacteremia among septic patients, which is important to physicians in emergency department. The study population is large, and the idea is good. However, there are some concerns before accepting it for publication.
1. As I know, logistic regression is also a method of machine learning. Why did author want to compare it with other machine learning methods? In fact, the authors conducted this study to compare different machine learning methods (LR, SVM and random forest). Maybe the title should be adjusted.
2. In methods, how did authors establish the models should be described in detail. Such as what software and what package were used? (Python? Scikit-learn?)
3. In building model with random forest, how many trees and its depth were applied?
4. Since the authors compared LR to SVM and RF, what’s the main difference between these three machine learning methods should be discussed, especially their advantages and disadvantages in such imbalance data set.
5. Can authors discuss the significance of deep learning, the most powerful machine learning algorithm? It learns from raw data input, as authors do have a large cohort, and with useful patterns enable accurate task decision making.
6. In results, could author provide the performance of different models with sensitivity, specificity, and positive likelihood ratio?
Reviewer 2 Report
Dear Authors,
I carefully read the article Predicting bacteremia among septic patients based on ED information by logistic regression and machine learning: A comparative study.
I have the following observations:
- the article is well written, but the Abstract is missing?!
- the conclusions are too brief, please redo the subsection and highlight the most important conclusions based on your results
- to figures 2 and 3, please add more details and explain the statistical details in the figures.
- table 2 is difficult to follow, I recommend turning it into a figure (figures) and adding a legend that explains in detail the statistical significance;
- please describe in more detail the mathematical and machine learning model.
- please correct the mistakes in English
Round 2
Reviewer 1 Report
Thanks for your revision. However, there are still some concerns.
About the Abstract
1. (This study aims to establish a predictive model for bacteremia in septic patients using available big data in the emergency department (ED) through logistic regression and machine learning (ML)) à suggest revise to “…logistic regression and other machine learning methods.”
2. (Both Model III and IV were derived from machine learning.) à suggest revise to “…derived from support vector machine (SVM) and random forest (RF).”
3. (The AUROC values of Models III and IV in the derivation group were 0.751 (95% CI, 0.740–0.761) and 0.835 (95% CI, 0.825–0.844) àThe result of derivation group in abstract seems to be redundant. Because after the model was established, we care more about its utility on unknown data. Suggest just keep the performance of validation dataset in abstract.
4. (There is no statistical evidence to support that the performance of the model created with logistic regression is superior to those created by ML.)à Please consider replaced by “SVM and RF”. Deep learning is included in ML, but authors did not test the performance of DL. In addition, there are other machine learning algorithms not tested by authors. Using “ML” here is inadequate.
5. (Discussion: We chose patients who met the previous Sepsis-2 criteria rather than Sepsis-3 because the revised definition was not suitable for ED physicians. These models could help medical staff identify patients at high risk and prevent unnecessary antibiotic use) à It’s discussion, is it necessary to show this in Abstract?
6. (Actually, the performance of ML was not inferior to that of logistic regression)à again, “SVM and RF was not inferior to…”,
About Discussion, in line 4:
7. Furthermore, the SVM and RF models predicted bacteremia and those established by LR. à Do authors mean “Furthermore, the SVM and RF models predicted bacteremia like those established by LR.” ?
